# Evolutionary Adaptation of Genes Involved in Galactose Derivatives Metabolism in Oil-Tea Specialized *Andrena* Species

**DOI:** 10.3390/genes14051117

**Published:** 2023-05-22

**Authors:** Gonghua Lin, Zuhao Huang, Bo He, Kai Jiang, Tianjuan Su, Fang Zhao

**Affiliations:** School of Life Sciences, Jinggangshan University, Ji’an 343009, China; lingonghua@163.com (G.L.); hzhow@163.com (Z.H.); hebo90@126.com (B.H.); jk_jxau@163.com (K.J.); sutianjuan126@126.com (T.S.)

**Keywords:** *A. camellia*, genome, RNASeq, molecular evolution, gene expression, galactose derivatives, NAGA-like

## Abstract

Oil-tea (*Camellia oleifera*) is a woody oil crop whose nectar includes galactose derivatives that are toxic to honey bees. Interestingly, some mining bees of the genus *Andrena* can entirely live on the nectar (and pollen) of oil-tea and are able to metabolize these galactose derivatives. We present the first next-generation genomes for five and one *Andrena* species that are, respectively, specialized and non-specialized oil-tea pollinators and, combining these with the published genomes of six other *Andrena* species which did not visit oil-tea, we performed molecular evolution analyses on the genes involved in the metabolizing of galactose derivatives. The six genes (*NAGA*, *NAGA-like*, *galM*, *galK*, *galT*, and *galE*) involved in galactose derivatives metabolism were identified in the five oil-tea specialized species, but only five (with the exception of *NAGA-like*) were discovered in the other *Andrena* species. Molecular evolution analyses revealed that *NAGA-like*, *galK*, and *galT* in oil-tea specialized species appeared under positive selection. RNASeq analyses showed that *NAGA-like*, *galK*, and *galT* were significantly up-regulated in the specialized pollinator *Andrena camellia* compared to the non-specialized pollinator *Andrena chekiangensis*. Our study demonstrated that the genes *NAGA-like*, *galK*, and *galT* have played an important role in the evolutionary adaptation of the oil-tea specialized *Andrena* species.

## 1. Introduction

Bees have a monophyletic lineage within a clade of Anthophila in the superfamily Apoidea [1]. There are around 20,000 different species of bees and, thanks to their effective pollination for both crops and natural flora, bees are now essential parts of practically all terrestrial ecosystems [2,3]. Bees can be broadly divided into two functional groups based on floral specificity: oligolectic species, which forage on one or a small number of closely related plant species, and polylectic species, which collect nectar and/or pollen from many different plant species [4]. The most well-known bee species include the polylectic honey bees (*Apis* spp.) and bumble bees (*Bombus* spp.). Numerous biological and ecological investigations of the polylectic species have been conducted [2,5]. In contrast, oligolectic bees have received significantly less research despite making up a sizable fraction of the world’s bee fauna [6]. For oligolectic bee exploitation and conservation, therefore, a greater understanding of their biology and ecology is required.

*Andrena* Fabricius (Andrenidae) is a large bee genus of around 1600 species with a wide distribution mainly throughout the Holarctic. They are a crucial pollinator in both natural and agricultural contexts, and they are a particularly important aspect of northern temperate ecosystems [7]. *Andrena* species exhibit a spectrum of diet breadth, from polylectic to oligolectic, which makes this genus a superb group to study the evolution of diet specialization [8,9]. Oil-tea (*C. oleifera*) is an important woody oil crop in many countries, including China, the Philippines, India, Brazil, and South Korea [10]. This plant only blooms in November and December [11], when there are few wild pollinators present. As a result, crop yields are very limited due to significant pollinator constraints [12,13]. Local farmers have attempted to utilize domestic honey bees (*Apis mellifera* and *Apis cerana*) to increase pollination efficiency, however, both species are badly damaged by the nectar’s toxicity [14]. Interestingly, some wild bees such as *A. camellia* and *Colletes gigas* primarily rely on oil-tea nectar to survive, suggesting these species have coevolved to become experts in oil-tea [15,16,17].

Direct observation on floral visiting and microscope examination of pollen from pollen baskets showed that *A. camellia* near exclusively collects nectar and pollen from oil-tea blossom [12]. *A. camellia* emerges in the middle of October and keeps its activities (mating, oviposition, and larval development) mainly during November and December, well synchronizing with the blossom period of oil-tea tree [11,12]. Besides *A. camellia*, at least three other *Andrena* bees (*A. chekiangensis*, *Andrena hunanensis*, and *Andrena striata*) are also reported to visit oil tea flowers [18,19]. *A. chekiangensis* is much larger than *A. camellia*, which makes it easier to identify between the two species. Our field observations found that, *A. chekiangensis* is strictly not an oil-tea specialist, because it also frequently visits tea tree blossoms (*Camellia sinensis*). In some sympatric habitats of *C. oleifera* and *C. sinensis*, *A. chekiangensis* individuals were more frequently observed in *C. sinensis* blossoms. On the contrary, *A. hunanensis* and *A. striata* were almost indistinguishable from *A. camellia* in terms of morphology features and feeding habits (oil-tea specialization). It should be noted that *A. hunanensis* and *A. striata* are frequently mistakenly classified as *A. camellia* in many amateur fieldwork reports. Due to the morphological similarity among *A. camellia* and its close relatives, the researchers believe there may even be hidden unknown species that have yet to be discovered.

Understanding the poisoning and detoxifying processes used by various bee species could be crucial to enhancing oil-tea flower pollination. According to chemical tests, the primary toxins affecting western honey bees are the galactose derivatives: raffinose, manninotriose, and stachyose [20,21]. Typically, these oligosaccharides must be broken down in two steps: first, the α-galactosidine linkages are hydrolyzed by galactosidases to produce sucrose and galactose and, then, the galactose molecules are converted to UDP-glucose via the Leloir route [22,23,24]. It should be noted that little is known about the genes involved in the metabolism of galactose derivates in *Andrena* species. Here, we present the first next-generation sequencing of *Andrena* species consuming oil-tea nectars. By combining these data with genomic information on other *Andrena* species from GenBank, we performed bioinformatic analyses of the genes involved in the metabolism of galactose derivatives. The objective is to determine whether the oil-tea-specialized *Andrena* species differ from the other *Andrena* species in terms of evolutionary specialization.

## 2. Materials and Methods

All pollinators were live-trapped from oil-tea blossoms in Jiangxi, Anhui, Sichuan, Zhejiang, Guangdong, and Hunan Provinces, China (Figure 1). In order to pick out *Andrena* samples, the specimens were identified based on morphological characteristics [18]. One female sample for each species was chosen for genome sequencing. Total genomic DNA was extracted from the thorax of each individual using the QIAGEN DNeasy Blood and Tissue kit (Germany), following the manufacturer’s protocols. DNA libraries with ~350 bp insertions were constructed and were then sequenced with both directions of 150 bp reads using the Illumina HiSeq 2000 sequencing platform (Illumina Inc., San Diego, CA, USA). Quality control for raw reads data was performed using fastp 0.20.0 with default settings and parameters [25].

The clean reads were used for de novo assembly with MEGAHIT [26]. The mitochondrial COI sequences were extracted by local blast using NCBI-BLAST+ program v2.13.0 [27] and were used as queries to search in BOLD system (www.boldsystems.org, accessed on 1 April 2023) to determine their taxonomic information. Previously published six *Andrena* genomes downloaded from GenBank were used as comparison objects: *Andrena dorsata*, GCA_929108735.1; *Andrena fulva*, GCA_946251845.1; *Andrena haemorrhoa*, GCA_910592295.1; *Andrena hattorfiana*, GCA_944738655.1; *Andrena minutula*, GCA_929113495.1; and *Andrena bucephala*, GCA_947577245.1. All 13 mitochondrial coding sequences in the genomes sequenced in this study and those from GenBank were extracted and were concatenated to reconstruct phylogenetic trees using IQ-TREE [28].

The galactose metabolism pathway of Hymenoptera in KEGG (ko00052) was used to identify the candidate genes involved in galactose derivatives metabolism. With honey bee (*A. mellifera*) candidate genes as query sequences, the exonerate program v2.4.0 [29] was used to find homologous genes in *Andrena* genomes. MEGA v10 [30] was used to combine and align the coding sequences from all samples for each gene. DNasP v6 [31] was used to identify the genetic variation information for each gene. The sequence similarity information among the genes as well as their translated protein sequences are calculated using clustal omega [32].

The molecular evolution analyses were performed in PAML program package [33]. The branch model was used to estimate the *dN/dS* (nonsynonymous/synonymous mutation) ratios of the foreground clade (oil-tea specialized species). The likelihood ratio tests (LRTs) between M0 (null model) and the branch model were performed by comparing twice the difference in log-likelihood values (*2*Δ*lnL*) against a chi-square distribution (*df* = 2). We also used the branch-site model called Model A to test for positive selections in the foreground clade oil-tea specialized species. The null model for Model A is Model A1, which is a modification of Model A, but with *ω2* = 1 fixed [34,35]. The putative positive selection sites were deduced with Bayes Empirical Bayes (BEB) analysis. It should be noted that the seven non-specialized *Andrena* species lacked *NAGA-like* gene. Due to the high sequence similarity between *NAGA* and *NAGA-like* genes, we arbitrarily used *NAGA* genes of these species instead.

In order to analyze the relative expression level of each gene, we also carried out RNAseq sequencing for *A. camellia* and *A. chekiangensis*, representing specialized and non-specialized oil-tea pollinators, respectively. Total mRNA was isolated from whole specimens of each individual (4 individuals were analyzed for each species). Following the manufacturer’s instructions, 150 bp reads were sequenced bidirectionally by the Illumina platform (Illumina, San Diego, CA, USA). The obtained clean reads of a randomly selected individual of each species were used for de novo assembly using the Trinity program [36]. The transcripts were processed by CD-HIT-EST [37] to remove the redundant sequences, and the generated unigenes were then used to predict coding sequences (CDSs) with the GeneMarkS-T program [38].

Orthologous genes were identified using OrthoFinder v2.3.11 [39]. The salmon program v1.0.0 [40] was used to calculate the expected read counts and transcripts per million (TPM) value for every orthologous gene, which were then used to identify differentially expressed genes (DEGs) with the DEBrowser program [41]. The genes with posterior fold changes (*FC*) in *A. camellia* against *A. chekiangensis* over two (i.e., *FC* > 2 or *FC* < 0.5) and with highly significant posterior probabilities of differential expression (*Padj* < 0.05) were considered to be DEGs. It should be noted that one of our candidate genes for the galactose derivatives metabolism had two closely related copies (*NAGA* and *NAGA-like*, see below), which made it challenging to pinpoint the true source of their mapped reads. In order to differentiate the relative expression levels between the copies, we initially extracted all reads that map to the two copies using bowtie2 program [42] and samtools [43]. We then randomly selected five 50 bp variable segments (in each segment, at least seven variable sites occurred between the two gene copies) as baits, and directly counted the number of reads that match the baits using grep module of seqkit program [44].

## 3. Results

Based on the morphological characteristics and DNA barcoding using mitochondrial COI sequences, six distinct *Andrena* species were discovered. Four species were recognized: *A. camellia*, *A. hunanensis*, *A. striata*, and *A. chekiangensis*. Since neither GenBank blasting nor BOLDSYSTEMS searching produced any COI hits for the two remaining species, they were temporarily designated as *Andrena* sp. 1 and *Andrena* sp. 2 (Table 1). A total of 62.74 giga bases (Gb) of WGS clean reads were obtained for the six *Andrena* species (five oil-tea-specialized species and *A. chekiangensis*). After assembly, 345~389 Mb of contigs were generated, with an N50 contig size of 8.8~15.6 Kb (Table 2). Phylogenetic analysis showed that the known three species (*A. camellia*, *A. hunanensis*, and *A. striata*) and the unknown two species formed to a single clade, while *A. chekiangensis* and *A. haemorrhoa* formed another clade (Figure 2).

According to the KEGG database, α-galactosidase (EC 3.2.1.22, also known as α-galactosidase A), which is prevalent in chordates, plants, and bacteria, is not found in arthropods such as bees (Hymenoptera) and other insects. As an alternative, bees have the equivalent α-N-acetylgalactosaminidase (*NAGA*, EC 3.2.1.49, also called α-galactosidase B), which is a homologous gene of α-galactosidase. Similar to other bees, there was only one *NAGA* gene in the *A. chekiangensis* genome. Intriguingly, two close, similar copies of the *NAGA* gene were discovered in the genome of the five oil-tea-specialized species. One was the conventional *NAGA*, while the other appeared to be a novel copy of the conventional *NAGA*. For ease of use, we refer to the novel copy as a *NAGA-like* gene. The *NAGA* and *NAGA-like* genes were highly similar; taking *A. camellia* as an example, there were 86% and 78% identity sites between them at nucleotide and amino acid sequence levels, respectively. It is worth noting that *NANA-like* had a termination mutation in the last exon (the sixth exon) which resulted in a shortened protein (Figure 3). All genomes of the five oil-tea-specialized species and the other seven species (including *A. chekiangensis*) contained four of the Leloir pathway genes [22], which most organisms use to metabolize galactose: *aldose 1-epimerase* (*galM*, EC 5.1.3.3), *galactokinase* (*galK*, EC 2.7.1.6), *galactose-1-phosphate uridylyltransferase* (*galT*, EC 2.7.7.12), and *UDP-galactose 4-epimerase* (*galE*, EC 5.1.3.2).

Genetic variations were surveyed within the five oil-tea-specialized species. The coding sequence of *NAGA* was 1320 bp in length, with 16 (1.21%) variable sites among the five species. The *NAGA-like* gene was shorter but, interspecifically, much more variable (2.66%) than *NAGA*. The four Leloir pathway genes were shorter than *NAGA* and *NAGA-like* and the number of variable sites ranked as *galM* > *galK* > *galT* > *galE* (Table 3). The sequences of the six genes of all the 12 *Andrena* species analyzed in this study are shown in the Appendix A. Branch model analyses were executed with the five oil-tea-specialized species as foreground clade and the remaining seven species as background clade. The results showed that *NAGA-like*, *galK*, and *galT* had significantly greater *dN*/*dS* ratios in the foreground clade than in the background clade (*χ*^2^ test, *df* = 1, *p* < 0.001). However, no significant divergence was seen for *NAGA*, *galM*, and *galE* (*p* > 0.2) (Table 4). We also assessed the putative positive selection sites in the two genes using the branch-site model test. With the Bayes Empirical Bayes (BEB) analysis, twelve sites in *NAGA-like*, four sites in *galK*, and one site in *galT* were found under positive selection with posterior probabilities >0.95.

A total of 33.7 Gb and 35.1 Gb of RNASeq clean reads were obtained for *A. camellia* and *A. chekiangensis*, respectively (Table 5). The assembly of *A. camellia* generated 23,087 unigenes, with a N50 value of 1668 bp. For *A. chekiangensis*, 18,387 unigenes were produced with a N50 value of 1692 bp. A total of 7151 orthologs were shared by the two species, with a total length of 7,847,865 bp and N50 of 1632 bp. The average value of relative expression level (TPM) of these orthologs in each sample was 140. The TPM values of the six candidate galactose metabolism genes are shown in Table 6. Taking the *A. chekiangensis* samples as the control group, 1987 differentially expressed genes were detected, including 1155 up-regulated (*FC* > 2) and 853 down-regulated (*FC* < 0.5) genes in *A. camellia*. *NAGA-like*, *galK*, and *galT* were significantly up-regulated in *A. camellia* (*FC* > 2, *Padj* < 0.05), while *NAGA*, *galM*, and *galE* did not deviate in expression levels between *A. camellia* and *A. chekiangensis* (*Padj* > 0.05) (Figure 4).

## 4. Discussion

Oil-tea is an important woody edible and industrial oil tree species [45]. Its product, tea oil, was categorized by the FAO (Food and Agriculture Organization of the United Nations) as a premium health-grade edible oil [46]. This plant presents a low oil yield because of self-incompatibility. Previous studies showed that the oil yield can be improved by an increase in pollinating insects [19,47]. However, blooms occur in late autumn and winter (from October to January), when bee pollinators are few due to cold temperatures. Additionally, some compounds in the nectar are toxic to most bees, including managed honey bees [13]. It is interesting to note that the poisonous elements in oil-tea nectar can be detoxified by both adults and larvae of several *Andrena* species. Unfortunately, despite the significant attention these species have received [47,48,49,50], little is known about the molecular mechanisms of detoxification. In this study, we presented the first genome and transcriptome sequencing of oil-tea specialized *Andrena* species and carried out bioinformatic analyses on the genes involved in galactose derivatives metabolism.

As stated in the introduction, in order to degrade galactose derivatives, the α-galactosidine bonds need to be first hydrolyzed to release galactose residues. In most organisms, such as chordates, plants, and bacteria, this process is accomplished by α-galactosidase A. However, honey bees and other insects have not yet been found to contain such a gene. Instead, *NAGA*, a homologous gene of *α-galactosidase A*, was commonly present in insect genomes. Although the protein produced by *NAGA* was initially thought to be an isozyme of α-galactosidase and given the name α-galactosidase B, it was actually an exoglycosidase acting on N-acetylgalactosamine [51]. There is no proof that it can replace α-galactosidase A’s role, which is why honey bees (such as *A. mellifera* and *A. cerana*) are unable to process oil-tea nectar. According to our molecular evolution analyses, there was no discernible selective differentiation of *NAGA* between the five oil-tea-specialized species and the other *Andrena* species. According to gene expression assessments, *NAGA* in *A. camellia* was not significantly deviated from that in *A. chekiangensis* (*p* > 0.05). As a result, we hypothesize that the conventional *NAGA* makes no contribution to the detoxification of oil-tea-specialized *Andrena* species.

The most intriguing discovery in this study might be the novel copy of *NAGA*, named the *NAGA-like* gene, in the five oil-tea-specialized *Andrena* species. Such a gene duplication pattern was not found in the other seven *Andrena* species, including *A. chekiangensis* which also consume oil-tea nectar, although not specifically. We also examined the genomes and transcriptomes of *C. gigas*, another crucial oil-tea pollinator [17,19], and no novel copy was found. Considering that the five oil-tea-specialized *Andrena* species formed a monophyletic group, we speculated that the *NAGA-like* gene was created from a particular gene-duplicating event that occurred in the common ancestor of these species. Molecular evolution analyses with branch models and branch-site models indicated that *NAGA-like* was under strong positive selections, a sign that a new phenotype of this gene would arise for these species [52]. The seqkit grep counts showed that the majority of reads mapping on *NAGA* were actually from *NAGA-like*. As a result, it is possible to estimate that the expression level of *NAGA-like* in *A. camellia* is ~120 times that of *NAGA* in the same species, or ~94 times that of *NAGA* in *A. chekiangensis*. Additionally, given that each of the 7151 orthologs had an average expression level (TPM) of 140, *NAGA-like* had a TPM that was around 496 times the average value. Such a high degree of *NAGA-like* expression suggests that it is essential for oil-tea specialization in *A. camellia*, and maybe for the other four oil-tea-specialized species. Since the novel *NAGA-like* gene was highly similar (86% DNA identity) to conventional *NAGA*, it was logical to assume that the *NAGA-like* protein could likewise catalyze N-acetylgalactosamine residue. In other words, although more research is required to confirm this idea, we propose that *NAGA-like* may have acquired a new role to break the α-galactosidine linkages from galactose derivatives.

There are four steps in the classic Leloir pathway. Firstly, β-d-galactose (natural galactose) is epimerized to α-d-galactose by *galM*. Secondly, α-d-galactose is phosphorylated to yield α-d-galactose 1-phosphate by *galK*. Thirdly, *galT* catalyzes the transfer of a UMP (uridine monophosphate) group from UDP-glucose (uridine diphosphate glucose) to galactose 1-phosphate, thereby generating glucose 1-phosphate and UDP-galactose. Finally, UDP-galactose is converted to UDP-glucose by *galE* [22]. Our results of the branch model and branch-site model tests showed that, *galK* and *galT*, but not *galM* nor *galE*, showed significantly larger *dN/dS* ratios in the five oil-tea-specialized species than in the background *Andrena* species. Moreover, the gene expression analyses showed that *galK* and *galT*, but not *galM* nor *galE*, was significantly more upregulated in *A. camellia* than in *A. chekiangensis*. These findings suggested that *galK* and *galT* may have been crucial in helping the oil-tea-specialized *Andrena* species adapt. We hypothesize that an improvement in catalytic efficiency may result from positive selection in *galK* and *galT* of the oil-tea-specialized species. In contrast, although *galM* and *galE* were also involved in galactose metabolism, no significant deviations in molecular evolution and gene expression were detected between *A. camellia* and *A. chekiangensis*, suggesting that the universal activity and quantity of these two epimerases are sufficient to deal with the catalytic demand in the oil-tea-specialized species.

## 5. Conclusions

Our study clearly demonstrated that the genes involved in galactose derivatives metabolism were crucial in the evolution of the oil-tea-specialized *Andrena* species. A novel *NAGA-like* gene was created to aid in the hydrolysis of the galactose residue from galactose derivatives, while the *galK* and *galT* genes were functionally improved to speed up the metabolism of the hydrolyzed galactose. It should be noted that, despite the fact that these species can handle poisonous oligosaccharides, their too-small population densities make it appear as though they are unable to meet the pollination needs of oil-tea. Our findings would provide insight into the poisoning and detoxifying processes of various bee species. We propose that the genetic engineering of relevant genes in cultivated species such as *A. mellifera* may finally assist in meeting the enormous pollination needs.

## Figures and Tables

**Figure 1 genes-14-01117-f001:**
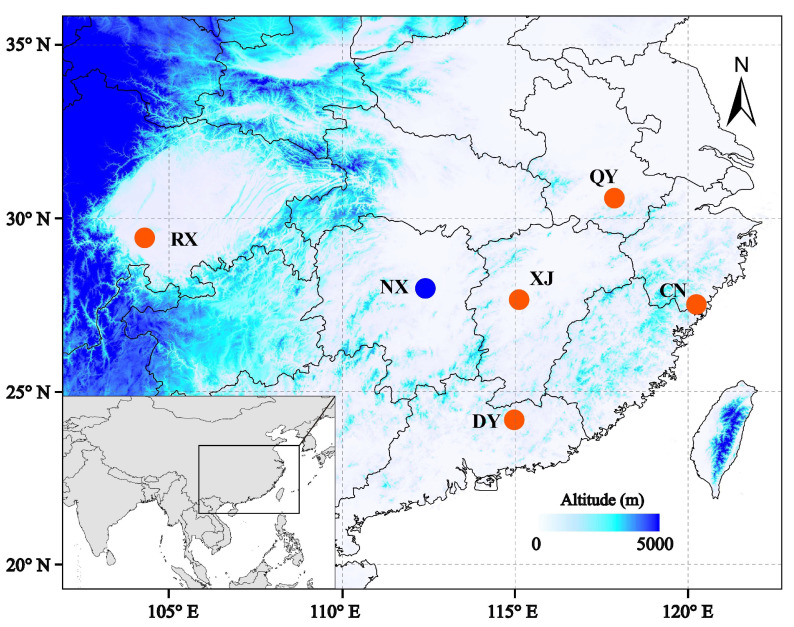
Sampling sites of the six *Andrena* species of pollinating oil-tea (red dots, oil-tea specialized species; blue dot, non-specialized oil-tea pollinator).

**Figure 2 genes-14-01117-f002:**
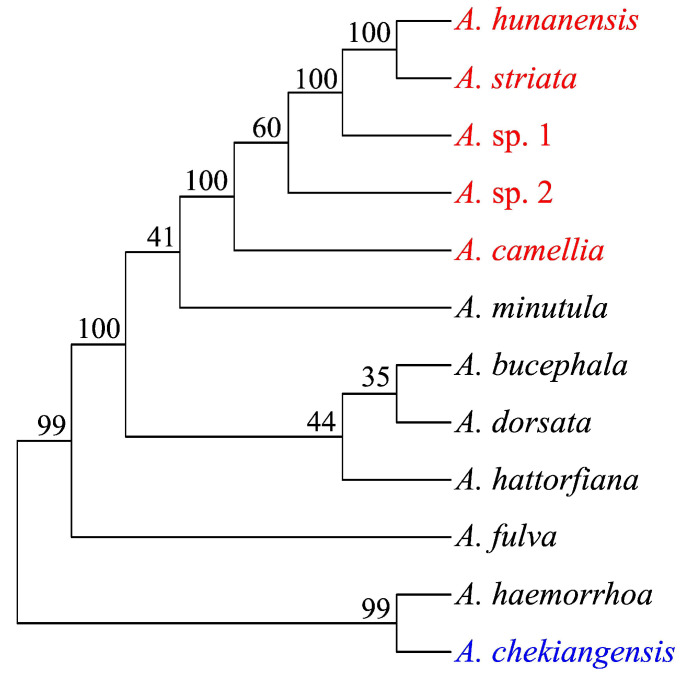
Phylogenetic relationship of 12 *Andrena* species analyzed in this study (red, specialized pollinator of oil-tea; blue, non-specialized pollinator of oil-tea; numbers beside each node represent percentages of bootstrap values).

**Figure 3 genes-14-01117-f003:**
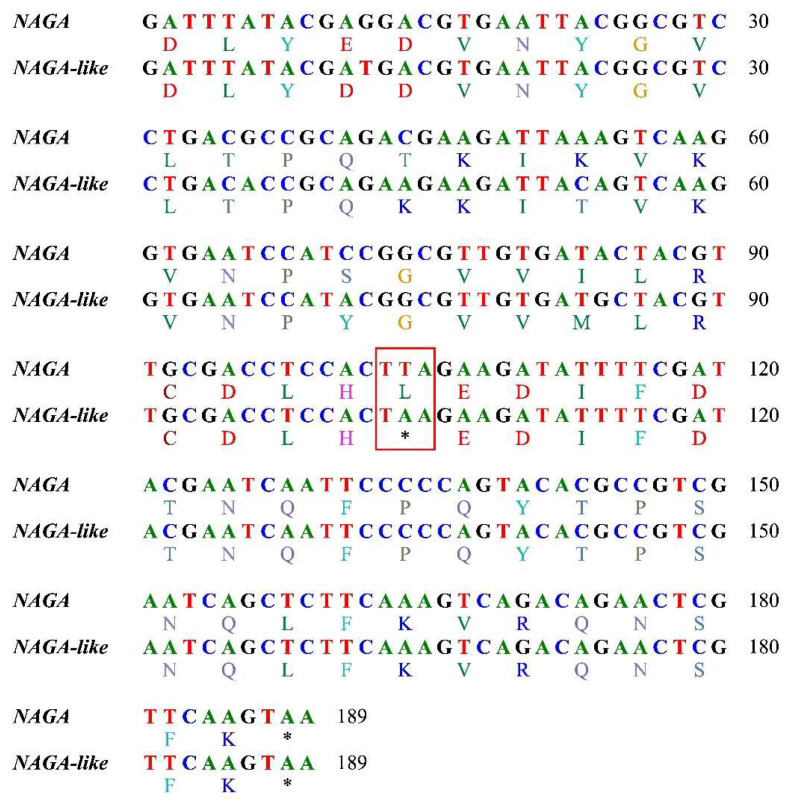
Alignment of nucleotide and amino acid sequences of the last exons of *NAGA* and *NAGA-like* genes of *A. camellia* (note in the red box the termination mutation in *NAGA-like*).

**Figure 4 genes-14-01117-f004:**
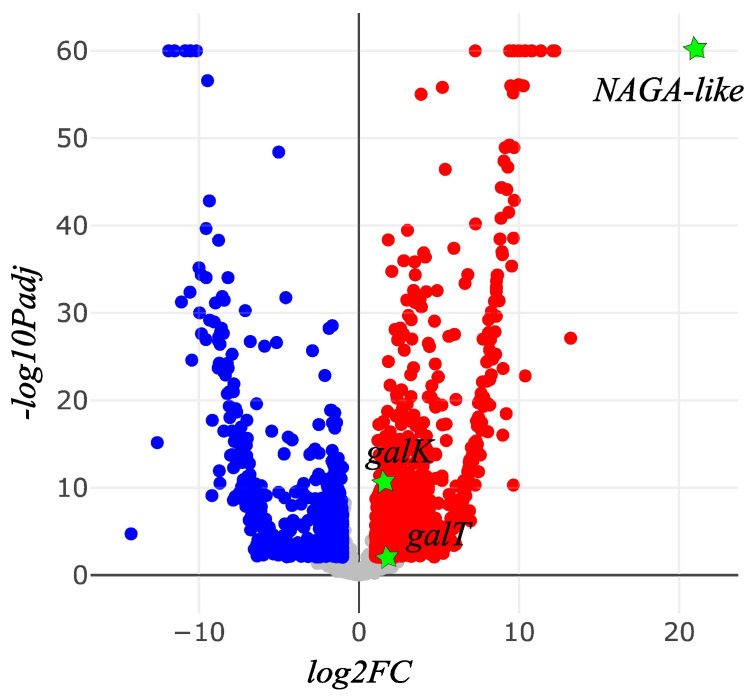
Differentially expressed genes of *A. camellia* against *A. chekiangensis* (blue dots, down-regulated; red dots, up-regulated; green stars, the three up-regulated genes involved in galactose derivatives metabolism).

**Table 1 genes-14-01117-t001:** Sample information of six *Andrena* species collected from oil-tea blossoms.

Sample	Species	Location	Longitude	Latitude
XJ01	*A. camellia*	Xiajiang, Jiangxi	115.1285	27.6546
QY01	*A. hunanensis*	Qingyang, Anhui	117.8796	30.5977
RX01	*A. striata*	Rongxian, Sichuan	104.2913	29.4377
CN02	*A.* sp. 1	Cangnan, Zhejiang	120.2556	27.4591
DY03	*A.* sp. 2	Dongyuan, Guangdong	114.9792	24.1905
NX04	*A. chekiangensis*	Ningxiang, Hunan	112.4206	27.9832

**Table 2 genes-14-01117-t002:** Short reads and assembly of next-generation genome of six *Andrena* species.

Species	Reads		Assembly	
	Length (Gb)	Accession	Length (Mb)	N50 (Kb)
*A. camellia*	10.38	SRR23869504	369.7	11.1
*A. hunanensis*	11.01	SRR23869503	384.3	8.2
*A. striata*	9.79	SRR23869502	393.1	8.5
*A.* sp. 1	10.75	SRR23869501	363.2	9.6
*A.* sp. 2	9.74	SRR23869500	353.5	14.7
*A. chekiangensis*	10.02	SRR23869499	365.7	11.6

**Table 3 genes-14-01117-t003:** Sequence length and genetic variation of genes involved in galactose derivatives metabolism within five oil-tea-specialized *Andrena* species.

Gene	Length	Variable Sites	Percent of Variable Sites
*NAGA*	1320	16	1.21
*NAGA-like*	1239	33	2.66
*galM*	1077	29	2.69
*galK*	1182	24	2.03
*galT*	1152	15	1.30
*galE*	1098	8	0.73

**Table 4 genes-14-01117-t004:** Branch model analyses of genes involved in galactose derivatives metabolism.

Gene	Foreground	Background	*2*Δ*lnL*	*p* (*df* = 1)
*NAGA*	0.049	0.023	1.412	0.234
*NAGA-like*	0.680	0.021	145.673	<1.000 × 10^−10^
*galM*	0.251	0.360	1.283	0.257
*galK*	0.864	0.161	24.279	8.336 × 10^−7^
*galT*	0.387	0.088	12.761	3.540 × 10^−4^
*galE*	0.129	0.063	0.913	0.339

**Table 5 genes-14-01117-t005:** RNASeq clean reads of *A. camellia* and *A. chekiangensis*.

Sample	Accession	Length (Gb)	Q30 (%)	GC (%)
Acam1	SRR8335252	7.54	91.55	45.84
Acam2	SRR8335251	9.50	92.41	46.35
Acam3	SRR8335254	7.64	91.76	46.23
Acam4	SRR8335253	9.04	92.35	45.97
Ache1	SRR23869498	9.03	95.82	41.95
Ache2	SRR23869497	8.88	96.10	42.46
Ache3	SRR23869496	7.82	96.16	41.61
Ache4	SRR23869495	9.36	95.81	43.69

**Table 6 genes-14-01117-t006:** Gene expression analyses of genes involved in galactose derivatives metabolism.

Gene	TPM (Mean ± SD)		*FC*	*Padj*
	*A. camellia*	*A. chekiangensis*		
*NAGA*	574.88 ± 162.33	740.45 ± 470.32	0.639	0.387
*NAGA-like*	69,465.93 ± 7690.83	0	+∞	−∞
*galM*	537.64 ± 78.69	210.82 ± 211.28	2.14	0.118
*galK*	172.87 ± 47.92	41.77 ± 34.05	3.35	0.016
*galT*	987.77 ± 139.04	318.55 ± 57.03	2.99	1.781 × 10^−12^
*galE*	32.45 ± 10.07	21.82 ± 14.73	1.23	0.624

## Data Availability

All data are presented in the text.

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
