# Peer review of "Evolutionary Adaptation of Genes Involved in Galactose Derivatives Metabolism in Oil-Tea Specialized Andrena Species"

_genes, 2023, doi:10.3390/genes14051117_

Round 1

Reviewer 1 Report

The article deals with a presentation for the first time of the genome and transcriptome sequencing on oil-tea specialized Andrena species.  The MS is sufficiently novel and interesting to warrant publication for the reason that it further carries out bioinformatic analyses on genes involved in galactose derivatives metabolism. It does add to the canon of knowledge and adheres to the standards of the journal «Genes».

The article is structured in the four following chapters: 1. Introduction; 2. Material and methods; 3. Results; 4. Discussion; 5. Conclusions. A reference list of fifty-five sources quoted in the main text is at the end. The «Introduction» summarizes the relevant research and provides context, explaining what other authors’ findings are being challenged or extended; here the bees are presented as a monophyletic lineage of the superfamily Apoidea and two functional groups are outlined: oligolectic and polylectic; the place of the genus Andrena in the system is outlined as well as its biological features and the connection with the oil-tea; it is of a satisfactory degree of completeness. The «Material and methods» include collection approaches across 6 provinces in China, DNA extraction, and molecular evolution analyses performance. The authors accurately explain how the data was collected and sampling is appropriate; the equipment and materials had been adequately described. The «Results» are presented concisely and dealt with the genetic variations within six oil-tea-specialized Andrena species. In addition, they are reinforced by six tables and three figures. The «Discussion» and the «Conclusions» are concise and consistent.

Further minor concerns:

Line 32

The “spp.” after Apis and Bombus not be in italic.

Line 155 & 156

The “sp.” after Andrena should not be in italic.

Line 160

“unknown” instead of “unknow”.

Line 210

A. camellia should be in italic.

Line 211

A. camellia & A. chekiangensis should be in italic.

Line 212 and 216-217

A. chekiangensis should be in italic.

Line 219

A. camellia (x2) should be in italic.

Line 221

A. camellia and A. chekiangensis should be in italic.

Line 283

The meaning of UMP should be explained.

Line 284 & 285

The meaning of UDP should be explained.

Line 289

               “and galK” instead of “and galT”?.

Line 292-303

               I am not convinced that the discussion part here is necessary. It should be avoided.

No comments.

Reviewer 2 Report

Dear Authors,

Your work raises the interesting problem of pollination of toxic plants by insects. 

I have one main comment and a few minor comments on your work.

1) You used total RNA to analyze gene expression. Is there a possibility that the isolated RNA (including the NAGA genes) does not belong to the bee, but to its microbiota (because bacteria also have this gene)?

Minor comments:

101 line: italicize species name

138 line: comma instead of dot ".. genes. which were .."

160 line: "unknow_" missing letter n

183-185 lines: italicize gene full names

210-221 lines: italicize gene full names and species names

Check the list of references, it must be issued uniformly and according to the rules of the journal
